# Changes in Soil Phosphorus Availability and Microbial Community Structures in Rhizospheres of Oilseed Rapes Induced by Intercropping with White Lupins

**DOI:** 10.3390/microorganisms11020326

**Published:** 2023-01-28

**Authors:** Siyu Chen, Da Yang, Yufei Wei, Lizhen He, Zujian Li, Shangdong Yang

**Affiliations:** Guangxi Key Laboratory of Agro-Environment and Agro-Products Safety, National Demonstration Center for Experimental Plant Science Education, Agricultural College, Guangxi University, Nanning 530004, China

**Keywords:** white lupin (*Lupinus albus* L.), oilseed rape (*Brassica campestris* L.), bacteria, rhizosphere, root exudates

## Abstract

Oilseed rape is sensitive to soil phosphorus deficiencies. In contrast, white lupin is widely used as a model plant because it has efficient phosphorus utilization. Therefore, soil fertility and microbial composition in the rhizospheres of oilseed rapes and root exudate metabolites were compared under monocropping and intercropping systems. The main purpose was to explore whether the phosphorus absorption of rapeseed can be promoted by intercropping with white lupine. In comparison with oilseed rape monoculture (RR), the results showed that the contents of soil-available phosphorus, microbial biomass and phosphorus in the rhizospheres of oilseed rapes in the intercropping system (RL) were all higher than those of RR. Meanwhile, in comparison with RR, not only phosphorus-solubilizing bacteria, such as *Streptomyces*, *Actinomadura* and *Bacillus*, but also phosphorus-solubilizing fungi, such as *Chaetomium*, *Aspergillus*, *Penicillium*, were enriched in the rhizospheres of the oilseed rape under the RL system. Moreover, more abundant soil bacterial functions, organic acids and metabolites were also detected in root exudates of the oilseed rapes under the RL system. All of the above results suggest that soil phosphorus availability in the rhizospheres of oilseed rape could be improved by intercropping with white lupin. Additionally, soil phosphorus-solubilizing microorganisms, that are enriched in the rhizospheres of oilseed rapes under RL systems, have an important function in the improvement of phosphorus absorption of rapeseed by intercropping with white lupin.

## 1. Introduction

Phosphorus deficiency greatly limits plant growth and leads to low yields [1]. Globally, approximately one-third of terrestrial soils do not have sufficient available phosphorus (AP) to optimize crop production. Many tropical acidic soils severely lack phosphorus [2,3]. Previous studies have also reported that as a nonrenewable resource, phosphorus resources will be exhausted by the end of this century [4]. Therefore, finding how to efficiently use phosphorus in agroecosystems is a major challenge in developing eco-agricultural industries [5].

Oilseed rape is very sensitive to phosphorus deficiencies, and its growth, development, yield and quality are significantly impacted by phosphorus deficiency [6,7,8,9]. In contrast, white lupin, as a plant that efficiently uses phosphorus, has excellent adaptability in phosphorus-deficient soil. First, it has cluster roots and exudes a large amount of phosphate mobilization substances, and even a small amount of AP can still be used by white lupin [10]. Second, white lupin in intercropping systems can lower soil pH, increase bacterial populations and secrete phosphorus-mobilizing substances, which are important factors that enable increased phosphorus uptake by maize and other crops in those systems [11,12,13]. In addition, many microorganisms can release soluble phosphorus through a variety of secretory mechanisms, such as acidification, ion exchange and organic acid production [14,15]. For example, phosphate-solubilizing microorganisms can improve plant phosphorus nutrition by converting soil-insoluble phosphorus into AP for plant growth [16,17]. Phosphorus-solubilizing microorganisms have been reported to increase phosphorus uptake and produce plant hormones. Meanwhile, phosphorus-solubilizing microorganisms also can fix nitrogen, enhance crop productivity and provide protection against plant pathogenic microorganisms [18,19,20]. In addition, root exudates can increase the solubility of insoluble phosphorus compounds in soil by secreting low-molecular-weight organic acids [21,22,23], which play an important role in improving the bioavailability of soil phosphorus [24]. The amount of root exudates is closely related to microbial biomass [25], as root exudates provide many nutrients and substantial energy for soil microorganisms, thus promoting soil microbial activity and playing an important role in microbial abundance and composition [26].

Currently, white lupin has been widely used as a model plant for efficient phosphorus utilization [27,28,29]. Moreover, white lupin can also provide phosphorus for neighboring plants, acting as an ecosystem promoter [30,31]. Studies have reported that the activity of soil acid phosphatase could be improved in an intercropping system of wheat/white lupin. In addition, white lupin has been shown to increase the absorption of phosphorus by wheat in wheat/white lupin intercropping systems [30].

Previous studies also found that P-efficient species/P-inefficient species inter-cropping could result in high yields without the use of phosphate fertilizer [32]. Intercropping refers to the mixed growth of two or more crops. This type of system can promote plants’ absorption for plant growth and convert resources into crop biomass [33,34,35]. In addition, the intercropping system significantly increased the yield of all intercropping species compared to those in the monoculture system. In particular, the total phosphorus uptake by crops in intercropping systems was higher than that of crops in monoculture systems [36,37]. In comparison with nonlegumes, legumes could convert organic phosphorus into inorganic phosphorus by producing enzymes, thus making more efficient use of phosphorus. Therefore, intercropping with legumes could effectively promote the acquisition of phosphorus in nonlegumes [24,33,36,37,38,39].

However, whether the phosphorus absorption of oilseed rape can be improved by intercropping with white lupins and its mechanisms are still unknown. We hypothesized that intercropping white lupin with rape could improve the soil phosphorus availability in the rhizospheres of rapes through root exudates and recruiting beneficial microorganisms. Therefore, the contents of soil AP, microbial compositions in rhizospheres and root exudate metabolites of oilseed rapes between monoculture and intercropping systems were analyzed.

## 2. Materials and Methods

### 2.1. Experimental Conditions and Plant Materials

The experimental trials were conducted in a large greenhouse (108°17′ E, 22°51′ N) at the vegetable base of the College of Agriculture, Guangxi University. The properties of the soil used in this experiment were as follows: pH 5.72, and the contents of total and available phosphorus were 0.64 g kg^−1^ and 16.49 mg kg^−1^, respectively.

The rape variety Nanyou 868 (R) and the white lupin variety Hualang (L) were used in this experiment. Rape and white lupin seeds were purchased from Fu Le Seedling Company, Sichuan and Man Mei Seedling Company, Jiangsu, respectively. In this experiment, three treatments were set up as follows, and each treatment was repeated twenty-five times: the soil used in the experiment without plants (CK), rape monoculture treatments (RR) and rape and white lupin intercropping treatments (RL). InRR, only rape plants grew in each bag. In RL, 1 rape plant grew surrounded by 4 white lupin plants in each bag. All treatments were managed under the same conditions.

On 4 February 2021, rape and white lupin seeds were sown evenly on the tray using peat mixture as a substrate. In March, after the rape plants entered the leaf development stage (BBCH codes: 14) [40], we transplanted them into 30 cm × 30 cm nonwoven bags. Samples were collected on 24 May 2021, when the rape entered the inflorescence emergence stage (BBCH codes: 50). A randomized block design was adopted in the experiment, with 3 treatments, each repeated 25 times. A total of 75 bags of seedlings were planted.

### 2.2. Soil Sampling Property Analysis

Soil samples were collected randomly from the roots of the rape and white lupin varieties mentioned above. A circle of soil with an approximately 25 cm radius was loosened with a sterilized shovel around the center of the plant, and then the whole plant was pulled up by hand by holding the base of the plant. The rhizospheric soils were collected using the shaking method [41]. Meanwhile, the root exudates were also collected [42]. Six single plants with the same growth characteristics were selected for each planting method; the bulk matrix of the plant roots was removed, and the impurities attached to the roots were washed with deionized water and sterile water in turn. Then, sterile gauze was used to remove moisture from the roots, and they were put into a light-shielding beaker filled with 200 mL of sterile water, and then ventilated continuously for 6 h. The crude extract containing root exudates was filtered immediately with a 0.22 μm filter (Corning Incorporated, Corning, NY, USA). The exudate samples were lyophilized using a Labconoco FreeZone 2.5 freeze dryer (Martin Christ, Osterode am Harz, Germany) and stored in a −80 °C refrigerator for root exudate analysis.

### 2.3. Analysis of Soil Available Phosphorus and Its Related Biological Properties

Soil AP content was measured using the 0.03 mol/L NH4F-0.025 mol/L HCl extract-stannous chloride glycerin solution colorimetric method. Soil samples were added to the extract solution, followed by shaking and filtering, and then we allowed the filtrate to absorb, added distilled water and ammonium molybdate reagent, mixed well, added stannous chloride glycerin solution and mixed well again, and used a 700 mm wavelength colorimetric method on a spectrophotometer [43].

Acid phosphatase activity in the soils was estimated by measuring the amount of ρNP released after incubating the samples with ρ-nitrophenyl-phosphate [18]. In a reaction tube, 0.25 mL of toluene, 4.0 mL of modified universal buffer (5× MUB, pH 6.0, which was made by dissolving 1.21 g of Tris, 1.16 g of maleic acid, 1.40 g of citric acid and 0.63 g of boric acid in 50 mL of 1 M NaOH to obtain a volume of 100 mL), and 1.0 mL ρN-nitrophenyl-phosphate (15 mmol L^−1^) were added to a 1.0 g soil sample and incubated at 30 °C for 1 h. The reaction was terminated by adding 1.0 mL of 0.5 mol CaCl_2_ and 4.0 mL of 0.5 mol NaOH to the mixture prior to filtration. The absorbance of the released ρNP was measured at 400 nm using a spectrophotometer (UV-1700, Shimadzu, Kyoto, Japan), and the phosphatase activity was expressed in mg ρ-NP g^−1^ h^−1^.

The soil microbial biomass P (MBP) was determined by means of the phosphorus molybdenum blue colorimetric method [44].

### 2.4. Analysis of Rhizosphere Microbial Diversity

Total DNA extraction, PCR amplification and sequence determination of the soil samples were performed by Shanghai Majorbio Biopharm Technology Co., Ltd. High-throughput sequencing was performed using the MiSeq platform.

Total DNA extraction was performed according to the instructions of the FastDNA^®^ Spin Kit for the rhizosphere (MP Biomedicals, Santa Ana, CA, USA), and the DNA concentration and purity were measured using a NanoDrop 2000 spectrophotometer (Thermo Fisher Scientific, Waltham, MA, USA). PCR amplification was performed on an ABI GeneAmp^®^ 9700 with the specific primers. PCR amplification of the 16S rRNA gene was as follows: initial denaturation at 95 °C for 3 min; then, 27 cycles were carried out: denaturation at 95 °C for 30 s, annealing at 55 °C for 30 s and extension at 72 °C for 45 s; then, extension at 72 °C for 10 min and ending at 4 °C. The ITS gene was amplified by PCR in the following ways: initial denaturation at 95 °C for 3 min, then 35 cycles, denaturation at 95 °C for 30 s, annealing at 53 °C for 30 s, extension at 72 °C for 45 s, followed by extension at 72 °C for 10 min and ending at 4 °C. The PCR analysis used TaKaRa rTaq DNA Polymerase.

The hypervariable region V3-V4 of the bacterial 16S rRNA gene was amplified with the bacterial primers 338F and 806R; while, the fungal ITS1 region was amplified with the primers ITS1F and ITS2R (ABI, Los Angeles, CA, USA) using ABI GeneAmp^®^ 9700 PCR thermal cycler and sequencing types shown in Table 1.

Illumina MiSeq sequencing: PCR products from the same sample were purified using the AxyPrep DNA Gel Extraction Kit (Axygen Biosciences, Union City, CA, USA), mixed and detected by recovery using a 2% agarose gel. The recovered products were quantified using a Quantus™ Fluorometer (Promega, Madison, WI, USA). Library construction was performed using the NEXTFLEX^®^ Rapid DNA-Seq Kit.

According to Majorbio Bio-Pharm Technology Co., Ltd., in the standard scheme, the purified amplifiers were combined with equal moles and double-ended sequencing (2 × 300) on the Illumina MiSeq platform (Illumina, San Diego, CA, USA). Sequencing was performed using Illumina’s MiSeq PE250 and MiSeq PE300 platform (Shanghai Majorbio Biopharm Technology Co., Ltd., Shanghai, China). The processing and analysis of sequencing data were described previously in detail [45,46]

### 2.5. Untargeted Metabolome Assays and Analysis

The 200 μL sample of root exudates was determined accurately and the 800 μL extract was added. After vortex mixing, the metabolites were extracted by low-temperature ultrasonic extraction. After centrifugation, the supernatant was removed, dried with nitrogen, re-dissolved with 120 μL complex solution, and centrifuged after low-temperature ultrasonic extraction to obtain the supernatant metabolite solvent for liquid-phase mass spectrometry analysis. Each sample was mixed with 20 μL of supernatant as a quality control (QC) sample. The volume of each QC was the same as that of the sample, and it was processed and tested in the same way as the analyzed sample. During the instrumental analysis, a QC sample was inserted into every 5–15 analyzed samples to examine the stability of the entire detection process. Semefeld’s ultra-performance liquid chromatography-tandem Fourier transform mass spectrometry (UHPLC-QprecisionHF-X) system was used for analysis. Chromatographic conditions were as follows: the chromatographic column was ACQUITYUPLCHSST3 (100 mm × 2.1 mm inner diameter, 1.8 m; Waters, Milford, CT, USA), mobile phase A was 95% water +5% acetonitrile (including 0.1% formic acid), mobile phase B was 47.5% acetonitrile +47.5% isopropanol +5% water (including 0.1% formic acid), the injection volume was 2 μL, the column temperature was 40 °C and the gradient of mobile phase is shown in Table 2. Data acquisition was carried out in full scanning mode, with the m/z range of 50–600. The metabonomics software ProGene-Sisqi (Waters Corporation, Milford, CT, USA) was used for peak extraction, comparison and identification, and finally a data matrix containing retention time, peak area, mass-to-charge ratio and identification information was obtained for post-processing and information analysis. Majorbio cloud platform 1 was used for multivariate analysis.

### 2.6. Untargeted Metabolome Assays and Analysis

The trial data were statistically analyzed using Excel 2019 and Statistical Product and Service Solutions (SPSS) 21, and the results are shown as the means with their standard deviations (mean ± SD). The calculation of alpha diversity of bacterial and fungal communities was performed using Mothur (v.1.30.2 https://mothur.org/wiki/calculators/, accessed on 2 August 2022). The Shannon and Insimpson indices were used to indicate the diversity of the rhizosphere soil microbial (bacteria and fungi) community, and the Ace and Chao1 indices were used to indicate the richness of the rhizosphere soil microbial community. Principal co-ordinate analysis (PCoA) and non-metric multidimensional scaling analysis (NMDS) were performed to evaluate the extent of the similarity of the endophytic microbial communities, and the R language (v.3.3.1) tool was used for statistical analysis and graphing. ANOSIM analysis was used to test the differences between groups, and Qiime was used to calculate the beta diversity distance matrix. The significance was based on 999 Monte Carlo permutations. Linear discriminant analysis (LDA) and the LDA effect size (LEfSe) method were used to identify significantly different microbial communities in the different environmental samples. The variance inflation factor (VIF) was used to screen environmental factors. Using RDA/CCA analysis to perform PCA analysis of environmental factor constraints, we analyzed the relationship between sample distribution and environmental factors, and reflected on the relationship among environmental factors, samples and flora. We used R (v.3.3.1) (pheatmap package) for correlation heatmap analysis to calculate the correlation between environmental factors and selected species. Online data analysis was performed using the free online platform of the Majorbio Cloud Platform (http://www.majorbio.com, accessed on 2 August 2022) of the Majorbio Bio-Pharm Technology Co. Ltd. (Shanghai, China).

## 3. Results

As shown in Figure 1, the contents of soil AP in the oilseed rape rhizosphere in the intercropping system were significantly higher than those in the monoculture system (Figure 1a). At the same time, the contents of soil AP in the oilseed rape rhizosphere in the monoculture system were not significantly different from those in the control (CK). The activities of soil phosphatase in the oilseed rape rhizospheres were not significantly different between the monoculture and intercropping systems. However, the oilseed rape rhizosphere in the monoculture system was significantly different from that in the CK (Figure 1b). However, the soil microbial biomass P (MBP) in the oilseed rape rhizospheres in the monoculture and intercropping systems was improved by cultivation with oilseed rape or lupin, but only intercropping with white lupin significantly promoted MBP compared with that in the CK (Figure 1c). This result suggested that the contents of soil AP and soil MBP in the oilseed rape rhizosphere could be improved by intercropping with white lupin.

As shown in Table 3, the indices of soil bacterial diversity in rhizospheres of oilseed rapes under the RR system were all significantly higher than those of the CK; however, there was no significant difference between RL and CK. Meanwhile, significant differences in soil bacterial richness (Ace and Chao1 indices) were not found in rhizospheres of oilseed rapes between the RR and RL systems.

In addition, the indices of soil fungal diversity in the RR system were all significantly higher than those of RL and CK. Additionally, the soil fungal richness showed the same trend as soil bacteria (Table 4).

Based on the Bray–Curtis distance in the principal coordinate analysis (PCoA), soil bacterial and fungal compositions were significantly divided into three groups, namely the RR and RL systems and CK, respectively (Figure 2a,c). Meanwhile, based on the Bray–Curtis distance in the nonmetric multidimensional scaling analysis (NMDS), soil bacterial and fungal compositions in the rhizospheres of oilseed rapes were significantly different between the RR and RL systems (Figure 2b,d).

As shown in Figure 3a, there were 10, 9 and 9 dominant soil bacterial phyla (i.e., relative abundances were greater than 1%) in the monoculture system (RR), intercropping system (RL) and control (CK), respectively.

In comparison with CK, Deinococcota, a dominant soil bacterial phylum, was lost in the oilseed rape rhizospheres under the RR and RL systems. Meanwhile, although the compositions of the dominant soil bacterial phyla in the oilseed rape rhizospheres did not significantly change, their proportions were altered between the RR and RL systems. In particular, the proportion of Actinobacteria (29.30%) and Firmicutes (10.12%) in the RL system obviously increased. This result suggests that the compositions of the dominant soil bacterial phyla did not change, but their proportions in the oilseed rape rhizospheres were altered by intercropping cultivation.

In addition, Ascomycota, unclassified_k__Fungi, Chytridiomycota, Mortierellomycota and Basidiomycota were the dominant soil fungal phyla (i.e., relative abundances were greater than 1%) in the RR, RL and CK (Figure 3b).

All of the above results show that the compositions of the dominant soil fungal phyla in the rhizospheres of oilseed rapes also did not change, but their proportions were altered by intercropping with white lupins. For example, in comparison with RR, the proportions of Ascomycota (86.99%), Chytridiomycota (1.65%), Actinobacteria (29.30%) and Firmicutes (10.12%) increased in RL.

As shown in Figure 4a, there were 21, 22 and 24 dominant soil bacterial genera (i.e., relative abundances were greater than 1%) among the CK, RR and RL, respectively.

In comparison with RR, the relative abundance of dominant soil bacterial genera in the rhizospheres of rapeseed rapes under the RL system were of *Streptomyces* (4.91%), *norank_f__JG30-KF-CM45* (2.78%), *Actinomadura* (2.52%), *Thermobifida* (2.33%), *Arthrobacter* (1.61%), *norank_f__norank_o__Vicinamibacterales* (1.57%), *norank_f__norank_o__norank_c__S0134_terrestrial_group* (1.50%), *Longispora* (1.43%), *norank_f__AKYG1722* (1.42%), respectively.

Among them, *OLB13* (1.34%) and *norank_f__BIrii41* (1.43%) were the unique dominant soil bacterial genera in the rhizospheres of oilseed rapes under the RR system; in contrast, *Hyphomicrobium* (1.04%) and *TM7a* (1.01%) were the unique dominant soil bacterial genera in the rhizospheres of oilseed rapes under the RL system.

In addition, at the genus level, there were 12 common dominant soil fungi genera in CK, RR and RL systems. They were *Humicola*, *Iodophanus*, *Mortierella*, *Fusarium*, *unclassified_f__Microascaceae*, *unclassified_f__Chaetomiaceae*, *Aspergillus*, *unclassified_c__Sordariomycetes*, *Arachniotus*, *unclassified_k__Fungi*, *Sodiomyces* and *Chaetomium*, respectively. However, the proportions of them were different. In comparison with RR, the proportions of *Chaetomium* (33.07%), *Aspergillus* (4.12%), *Penicillium* (1.65%), *Gibellulopsis* (1.51%), *Iodophanus (1*.44%) in the rhizospheres of oilseed rapes under the RL system were higher than those under the RR system (Figure 4b).

As shown in Figure 5a, there were 12, 40 and 33 unique bacteria in the oilseed rape rhizo-spheres in the RR and RL system and CK at the genus level, respectively. Furthermore, there were 46, 77 and 80 unique bacteria in the oilseed rape rhizospheres in the RR, RL system and CK at the OTU level, respectively (Figure 5b).

As seen in Figure 5c, there were 12, 40 and 33 unique bacterial genera in the oilseed rape rhizospheres in the RR, RL and CK, respectively (Figure 5c). Meanwhile, there were 46, 77 and 80 unique fungi in the oilseed rape rhizospheres in the RR, RL and CK at the OTUs level, respectively (Figure 5d).

Significant differences in soil bacteria in the oilseed rape rhizospheres among the monoculture (RR), and intercropping systems (RL) and CK were analyzed by means of the LEfSe method (LDA threshold of 3) (Figure 6a).

Based on the LEfSe analysis results, in comparison with the RR, Streptomyces were enriched in the rhizospheres of the RL. In contrast, norank_f__JG30-KF-AS9, Sphingomonas, Altererythrobacter, Micropepsis, Ellin6067, norank_f__BIrii41 and Gemmatimonas were enriched in the rhizospheres of the RR. Moreover, the number of dominant soil fungal groups in RL was similar to that in the RR system. For example, Arcopilus, Arnium, Penicillium and Goffeauzyma were enriched in the RL; whereas, Plectosphaerella, Dirkmeia and Ceratorhiza were enriched in the rhizospheres of the RR (Figure 6b).

Variance inflation factor (VIF) analysis was used to determine the collinearity among different factors. Soil pH had strong collinearity with other edaphic factors, such as soil water content (SWC, VIF = 6.73), phosphatase (VIF = 5.53), microbial biomass phosphorus (MBP, VIF = 6.47) and available phosphorus (AP, VIF = 7.56). Therefore, we separated soil pH from the other edaphic factors as environmental factors with *p* > 0.05 or VIF > 20. The screened environmental factors (SWC, phosphatase, MBP and AP) were used for the redundancy analysis (RDA) (Figure 7).

First, for soil bacteria, at the phylum level, the RDA showed an interpretability of 56.27% (RDA1: 40.62%; RDA2: 15.65%) (Figure 7a), while at the genus level, the RDA showed an interpretability of 45.47% (RDA1: 26.06%; RDA2: 19.41%) (Figure 7b). In general, the two axes explained most of the information about the soil’s biological and physicochemical properties and soil bacteria at the genus level. Meanwhile, the three treatments were obviously clustered into three circles, which showed significant differences in the response of the soil bacteria in rhizospheres of oilseed rape to AP, SWC, MBP and phosphatase among the three different treatments (*p* < 0.05). These results also suggest that soil bacteria at the phylum or genus level in the rhizospheres of oilseed rapes under different systems are affected by soil AP contents. (Figure 7a,b).

Second, for soil fungi, at the phylum level, the RDA showed an interpretability of 35.35% (RDA1: 19.92%; RDA2: 15.43%) (Figure 7c); at the genus level, the RDA showed an interpretability of 60.62% (RDA1: 48.34%; RDA2: 12.28%) (Figure 7d). Overall, the two axes explained most of the information on genus-level fungi, soil biological traits and physicochemical properties. Meanwhile, the three treatments were obviously clustered into three circles in the different quadrants, and they were distributed at three different positions of RD1. This indicated that the soil fungi at the genus level in the rhizospheres of oilseed rapes under different treatments were also affected by soil AP content. In particular, SWC, MBP and phosphatase responses were significantly different (Figure 7c,d).

Additionally, regarding the relationships between soil properties and bacteria, at the phylum level, SWC, MBP and phosphatase responded significantly differently. For example, SWC was significantly negatively correlated with Proteobacteria, negatively correlated with Bacteroidota and positively correlated with Halanaerobiaeota. Meanwhile, a positive correlation existed between phosphatase and Planctomycetota; whereas, a negative correlation was observed between MBP and Nitrospirota and Chloroflexi. Moreover, a significantly positive correlation between AP and Halanaerobiaeota and a very significantly negative correlation between AP and Armatimonadota were found (Figure 8a).

Similarly, regarding soil fungi, at the phylum level, SWC was significantly negatively correlated with Olpidiomycota. Meanwhile, soil microbial biomass phosphorus (MBP) demonstrated a positive correlation with Glomeromycota and Zoopagomycota and a very significantly negative correlation with Rozellomycota. Moreover, for the content of available phosphorus (AP), a significantly negative correlation with Mortierellomycota and a very significant negative correlation with Armatimonadota were also found (Figure 8c).

At the genus level, SWC, phosphatase, MBP and AP showed significant effects on soil bacterial community composition. (*p* < 0.05). First, SWC showed an extremely significant positive correlation with *Chaetomium* and *Arachniotus*, and a negative correlation with *Plectosphaerella*; second, phosphatase also showed a significant positive correlation with *unclassified_c__Sordariomycetes*, *unclassified_f__Chaetomiaceae*, *Pseudogymnoascus* and Gibellulopsis; third, MBP demonstrated a significant negative correlation with *Talaromyces*; Additionally, the content of soil available phosphorus (AP) also showed a significant negative correlation with *Neocosmospora*, *Mortierella*, *Humicola*, *Talaromyces*. Meanwhile, AP displayed an extremely significant positive correlation with *Penicillium*, *Pseudogymnoascus* and *unclassified_c__Sordariomycetes* (Figure 8d).

As shown in Figure 9, the root exudates of the oilseed rape between the RR and RL systems were largely separated. This result indicated that the metabolites of the oilseed rape root exudates were quite different. Moreover, the differences in the oilseed rape root metabolites between the RR and RL systems were also analyzed by an orthogonal partial least square discriminant analysis (OPLS-DA). The positive and negative data showed significant differences in the anions and cations in the oilseed rape root exudates between the RR and RL systems (Figure 9a–d).

In addition, the contents of 23 of the 42 rape root exudates were significantly higher in the RL system than in the RR system. Among them, the contents of five organic acids and derivatives; one organic oxygen compound; three organic nitrogen compounds; three organoheterocyclic compounds; one phenylpropanoid and polyketide; one nucleoside, nucleotide and analogue; two lipids and lipid-like molecules; and two benzenoid compounds in the oilseed rape root exudates in the RL system were significantly higher than those in the oilseed rape root exudates in the RR system (Table 5, Appendix A).

In addition, the secondary metabolic pathways of the KEGG in the oilseed rape root exudates in the RR and RL systems were amino acid metabolism, biosynthesis of other secondary metabolites, carbohydrate metabolism, lipid metabolism and metabolism of terpenoids and polyketides (Figure 10a). In addition, glycine, serine and threonine metabolism; the citrate cycle (TCA cycle); alanine, aspartate and glutamate metabolism; zeatin biosynthesis; and fatty acid elongation in the oilseed rape root exudates are the main metabolic pathways of rape root exudates, among which alanine, aspartate and glutamate metabolism in the RL system was significantly enriched compared with the oilseed rape root exudates in the RR system (Figure 10b,c).

The correlation between the top 10 most abundant rhizospheric bacteria at genus level and metabolites was also analyzed using the Spearman correlation algorithm and the Bray–Curtis distance algorithm.

In comparison with RR, *Streptomyces*, *Actinomadura*, *Thermobifida*, *Bacillus* and *Longispora* were the dominant soil bacterial genera in the rhizospheres of oilseed rapes under the RL system.

In addition, *Streptomyces* was negatively associated with lipids and lipid-like molecules (aucubin) and organic acids and derivatives (tryptophyl-cysteine), and it was positively associated with organic oxygen compounds (aminoacetone), 2-dodecylbenzenesulfonic acid and benzenoids (2-dodecylbenzenesulfonic acid). Meanwhile, *Actinomadura* was positively associated with organic oxygen compounds (aminoacetone); *Thermobifida* was positively associated with nucleosides, nucleotides and their analogues (lamivudine sulfoxide); organic oxygen compounds (aminoacetone); benzenoids (2-Dodecylbenzenesulfonic acid); organic nitrogen compounds (MEMANTINE); and lipids and lipid-like molecules (4,5-dihydrovomifoliol). Additionally, *Bacillus* was positively associated with nucleosides, nucleotides and their analogues (Lamivudine sulfoxide). *Longispora* was negatively associated with organic acids and derivatives (L-trans-5-hydroxy-2-piperidinecarboxylic acid, aucubin and tryptophyl-Cysteine), and it was positively associated with organic oxygen compounds (aminoacetone); benzenoids (2-dodecylbenzenesulfonic acid), organoheterocyclic compounds (cis-quinceoxepane), organic nitrogen compounds (MEMANTINE) and lipid-like molecules (4,5-dihydrovomifoliol) (Figure 11a).

Similarly, the correction between the top 10 most abundant rhizospheric fungi at the genus level and metabolites was also analyzed. In comparison with RR, *Chaetomium*, *Aspergillus* and *Penicillium* were the dominant soil fungal genera in the rhizospheres of oilseed rapes under the RL system.

Among them, *Chaetomium* was negatively associated with lipids, lipid-like molecules (Aucubin), organic acids and derivatives (Tryptophyl-Cysteine), and it was also positively associated with organic oxygen compounds (Aminoacetone) and benzenoids (2-Dodecylbenzenesulfonic acid). Meanwhile, *Aspergillus* was negatively associated with nucleosides, nucleotides, and their analogues (lamivudine sulfoxide) and organic acids and their derivatives (Leucyl-Threonine). *Penicillium* was positively associated with benzenoids (4-hydroxy-5-phenyltetrahydro-1,3-oxazin-2-one) (Figure 11b).

## 4. Discussion

Previous studies confirmed that white lupin is a genotype crop with high phosphorus utilization [47,48,49], and that white lupin can be used for activating phosphorus in the soil by means of its implementation in RL with other crops [6,50,51].

Moreover, soil microbial biomass is also an important indicator for evaluating soil fertility [46], and the turnover of soil MBP could release inorganic phosphorus [52]. Additionally, soil phosphatases can be produced by plant roots and microbes in soil [53]. In this paper, the soil AP contents in rhizospheres of oilseed rapes under the RL system were significantly higher than those of oilseed rapes under the RR system. Additionally, soil MBP in the rhizospheres of oilseed rapes under the RL system was significantly higher than that of oilseed rapes under the RR system. This result suggested that oilseed rape, a sensitive phosphorus-deficient crop, could be grown in phosphorus-deficient soils by intercropping with white lupin.

Soil quality is closely related to the population and composition of soil microorganisms, and soil microorganisms are ideal bio-indicators of soil health [54]. In comparison with the RR system, although soil bacterial diversities were not significantly improved, soil bacterial compositions in the rhizospheres of the oilseed rapes under the RL system were significantly altered and their functions were also significantly changed by intercropping with white lupin. Additionally, our results were similar to those of some other studies [55,56,57].

Previous studies confirmed that Actinobacteria could produce gluconic acid to dissolve organic phosphorus, and it was also a source of phosphatase [47]. Meanwhile, *Bacillus* [19,58,59] and *Streptomyces* [60] can promote plant growth by means of phosphorus solubilization and mobilization [61]. *Actinomadura* is positively correlated with AP [62]. In comparison with those in rhizospheres of oilseed rapes under the RR system, *Streptomyces*, *Actinomadura* and *Bacillus* were all enriched in the rhizospheres of oilseed rapes under the RL system. Additionally, the proportion of Actinobacteria was the most abundant soil bacterial phylum in rhizospheres of oilseed rapes under the RL system. The fact that phosphorus absorption and utilization by oilseed rapes could be promoted under the RL system is an important factor to consider.

Moreover, as a positive correlation between Ascomycota and available phosphorus could be found [63] and Chytridiomycota could utilize insoluble phosphorus [64], *Chaetomium* [65] could release phosphatase and phytase. *Aspergillus* [66] and Penicillium [67] are typical phosphate-solubilizing fungi. In our study, the proportions of Ascomycota, Chytridiomycota, *Chaetomium*, *Aspergillus*, *Penicillium* in the rhizospheres of oilseed rapes under the RL system were all higher than those of oilseed rapes under the RR system. Furthermore, *Streptomyces* and *Penicillium* were expressed significantly under the RL system according to LEfSe analysis.

All of the above results suggest that soil phosphorus-solubilizing microorganisms, such as *Bacillus*, *Streptomyces*, *Actinomadura*, *Chaetomium*, *Aspergillus*, *Penicillium* were enriched in rhizospheres of oilseed rapes under the RL system. Additionally, this is one of the important reasons for higher contents of soil AP being detected in the rhizospheres of oilseed rapes under the RL system.

In addition, citric acid has a strong ability to activate soil phosphorus [68]. Under phosphorus deficiency, plants usually secrete low-molecular-weight organic acids to activate soil phosphorus for improving phosphorus utilization [69]. The content of citric acid in the oilseed rape root exudates under the RL system was significantly lower than that under the RR system [70], but the total amount of metabolites in the root exudates, particularly the content of organic acids, in the oilseed rape root exudates was significantly lower under the RL system than that under the RR system. Moreover, P is involved in the synthesis of phosphocreatine, a high-energy P compound. Phosphate creatine hydrolysis releases a large amount of energy, regulates the activity of glycolytic enzymes and accelerates metabolic activity [71]. Our results also showed that significantly abundant creatine was also detected in the root exudates of oilseed rapes under the RL system compared to those of oilseed rapes under the RR system.

Root exudates have a direct effect on soil bacteria in the rhizosphere and can provide substrates for microbial growth and stimulate different microbial groups [72]. Most of the metabolites of the bacterial communities in rhizospheres and the root exudate metabolites under different rape cultivation systems contained two types of lipids and lipid-like molecules; two species of organic acids and derivatives; one species of organic oxygen compound; organic nitrogen compounds; benzenoids; and nucleosides, nucleotides and their analogs. These root exudate metabolites can improve with a higher content of soil AP. The enrichment of phosphorus-solubilizing microorganisms in rhizospheres and the different root exudates induced by intercropping with white lupin are the main factors for improving the soil AP contents in the rhizospheres of oilseed rapes under the RL system.

## 5. Conclusions

In comparison with the oilseed rape monoculture system, higher contents of soil AP and more abundant soil MBP in the rhizospheres of oilseed rapes were found under the oilseed rape/white lupin intercropping system. Meanwhile, the enrichment of phosphorus-solubilizing bacteria and fungi, such as *Streptomyces*, *Actinomadura* and *Bacillus*, *Chaetomium*, *Aspergillus* and *Penicillium*, could be detected in the rhizospheres of oilseed rapes under the oilseed rape/white lupin intercropping system. Moreover, the organic acids and 23 metabolites in the root exudates of the oilseed rapes under the oilseed rape/white lupin intercropping system were significantly different. Most of these organic acids and metabolites were positively correlated significantly with phosphate-solubilizing microorganisms. The enrichment of phosphorus-solubilizing microbes in rhizospheres of oilseed rapes was an important reason for the higher phosphorus absorption by oilseed rapes under the oilseed rape/white lupin intercropping system.

## Figures and Tables

**Figure 1 microorganisms-11-00326-f001:**
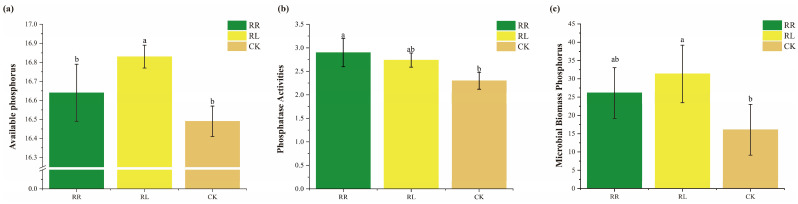
Available phosphorus concentrations (**a**), soil phosphatase activities (**b**) and soil microbial biomass phosphorus (MBP) (**c**) in the RR, RL and CK. All data are presented as the mean ± SD (standard deviation). Different letters in the column indicate significant differences among treatments at *p* < 0.05.

**Figure 2 microorganisms-11-00326-f002:**
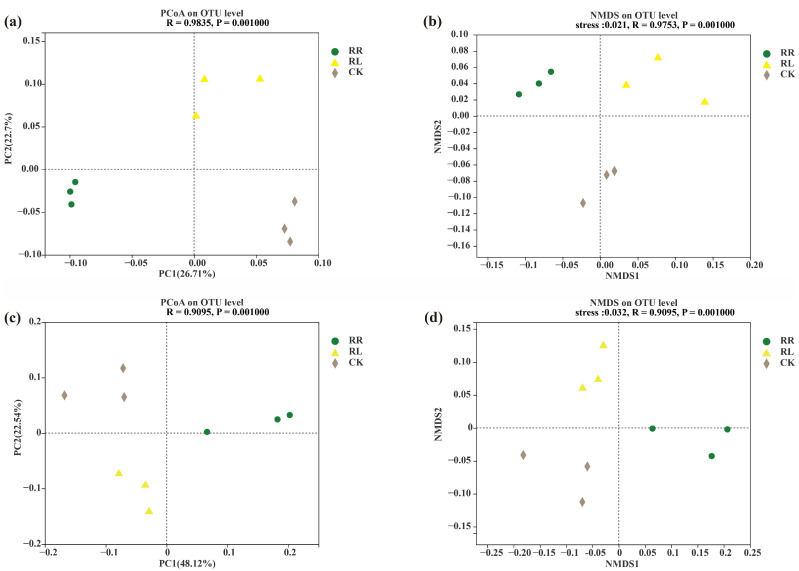
Comparison of soil microorganisms in rhizospheres of oilseed rapes in the monoculture (RR), intercropping system (RL) and background (CK); (**a**) PCoA of soil bacterial communities at the OTU level; (**b**) NMDS of soil bacterial communities at the OTU level. (**c**) PCoA of soil fungal communities at the OTU level; (**d**) NMDS of soil fungal communities at the OTU level.

**Figure 3 microorganisms-11-00326-f003:**
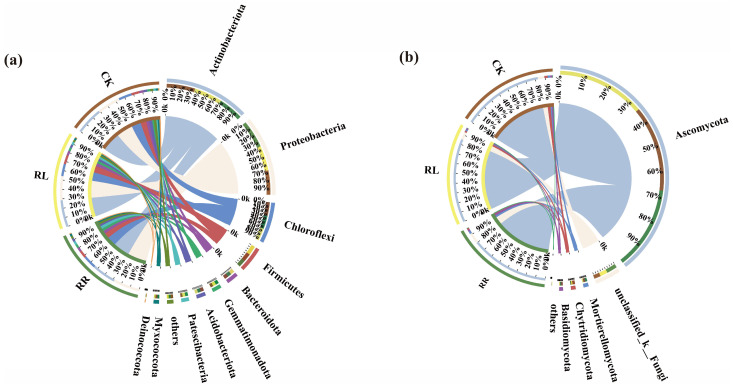
Distribution of the dominant soil bacterial (**a**) and fungal (**b**) phyla in rhizospheres of the oilseed rapes under the RR, RL and CK systems.

**Figure 4 microorganisms-11-00326-f004:**
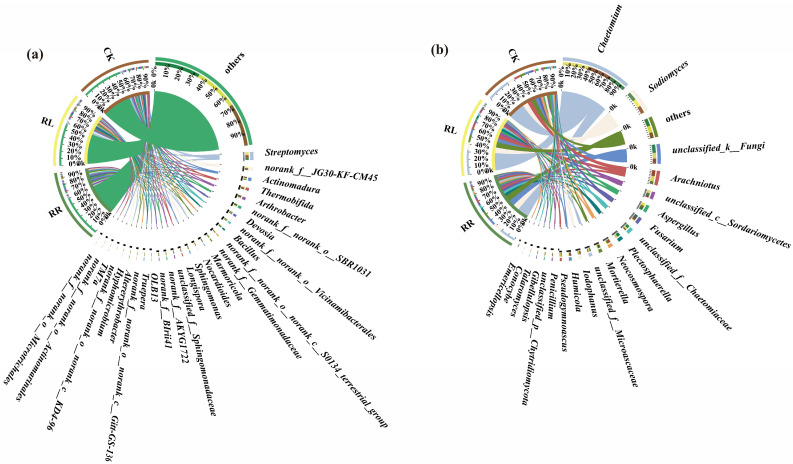
Proportions of the dominant soil bacterial (**a**) and fungal (**b**) genera in the oilseed rape rhizospheres in the monoculture system (RR), intercropping system (RL) and CK.

**Figure 5 microorganisms-11-00326-f005:**
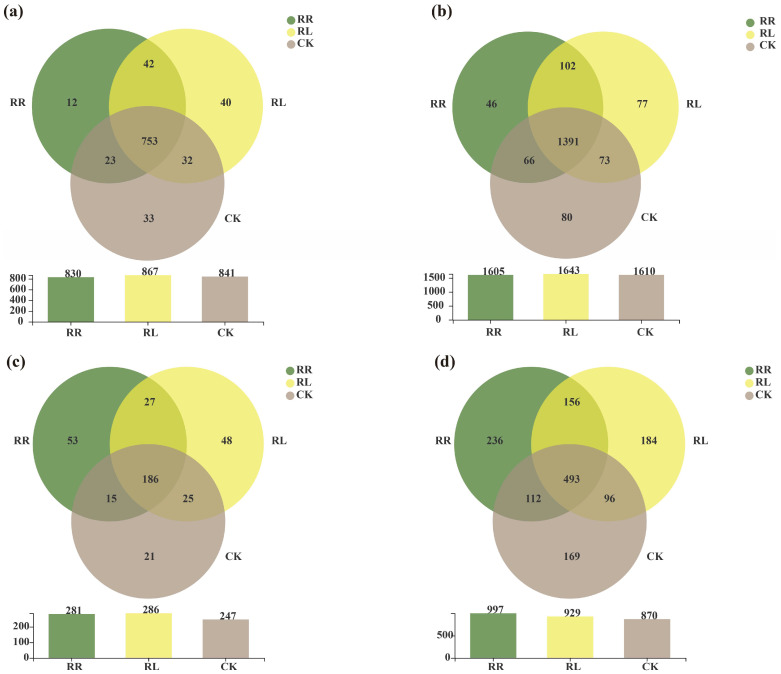
Venn diagram of soil bacteria in the oilseed rape rhizospheres in the monoculture (RR), intercropping system (RL) and CK. (**a**) Venn diagram of soil bacterial communities at the genus level; (**b**) Venn diagram of soil bacterial communities at the OTU level; (**c**) Venn of soil fungal communities at the genus level; (**d**) Venn diagram of soil fungal communities at the OTU level.

**Figure 6 microorganisms-11-00326-f006:**
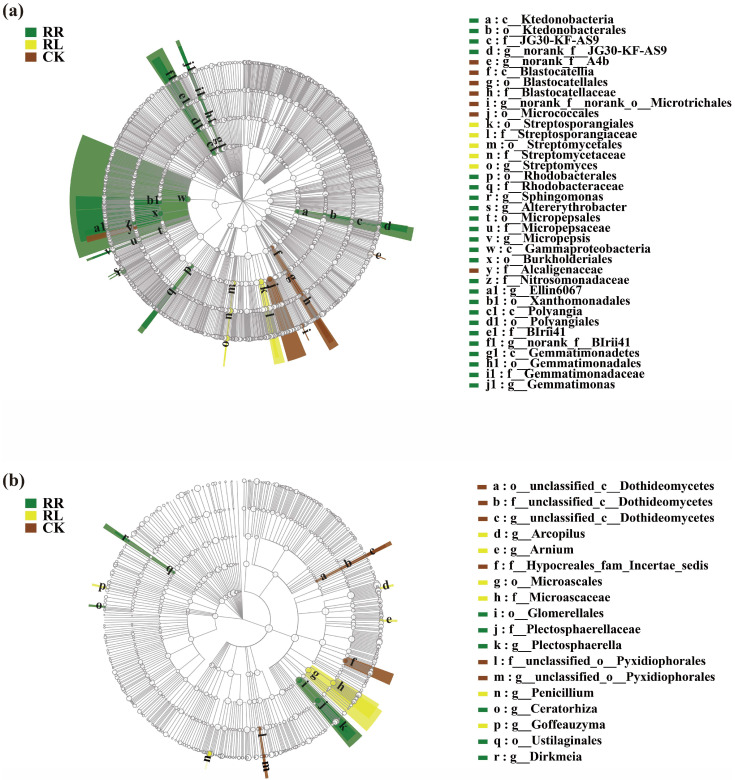
LEfSe analysis of soil bacteria (**a**) and fungi (**b**) in the rhizosphere of crops under the monoculture (RR) and intercropping system (RL) and CK. Different color regions represent different constituents (green: RR; yellow, RL; brown: CK). Circles indicate the phylogenetic level from the phylum to the genus. The diameter of each circle is proportional to the abundance of the group. Different prefixes indicate different levels (p: phylum; c: class, o: order; f: family; g: genus).

**Figure 7 microorganisms-11-00326-f007:**
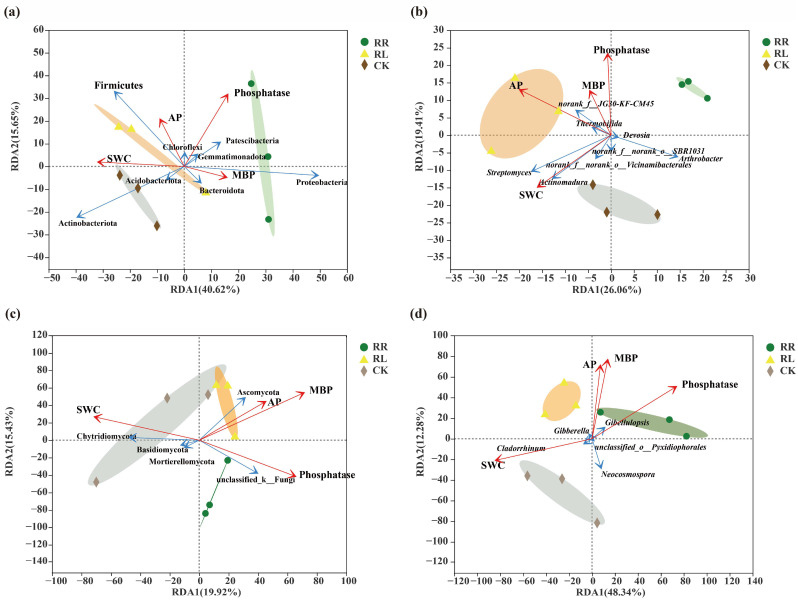
Redundancy analysis (RDA) of oilseed rape in the monoculture (RR), intercropping system (RL) and CK. SWC, soil water content; MBP, microbial biomass phosphorus; AP, available phosphorus. (**a**) Soil bacteria at the phylum level. (**b**) Soil bacteria at the genus level. (**c**) Soil fungi at the phylum level. (**d**) Soil fungi at the genus level.

**Figure 8 microorganisms-11-00326-f008:**
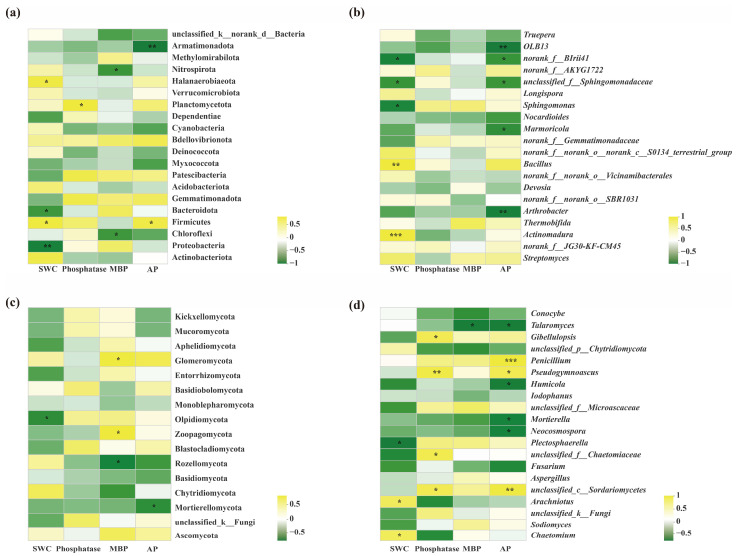
Heatmap correlation analysis between edaphic physicochemical factors and the bacterial population of oilseed rape in the RR, RL and CK. (**a**) Soil bacterial population at the phylum level; (**b**) Soil bacterial population at the genus level; (**c**) Soil fungal population at the phylum level; (**d**) Soil fungal population at the genus level. R values are shown in the figure with shades of color, and they are marked with *, ** or *** if the *p* values are less than 0.05, 0.01, 0.001, respectively. SWC, soil water content; MBP, microbial biomass phosphorus; AP, available phosphorus.At the genus level, only SWC and AP had significant effects on soil bacterial community compositions., i.e., SWC showed a significant positive correlation with *Actinomadura* and *Bacillus*. Meanwhile, SWC also showed a significant negative correlation with *Sphingomonas*, *unclassified_f__Sphingomonadaceae* and *norank_f__BIrii41*. Furthermore, AP also showed a significant negative correlation with *norank_f__BIrii41*, *unclassified_f__Sphingomonadaceae*, and *Marmoricola*, and an extremely significant negative correlation with *OLB13* and *Arthrobacter* (Figure 8b).

**Figure 9 microorganisms-11-00326-f009:**
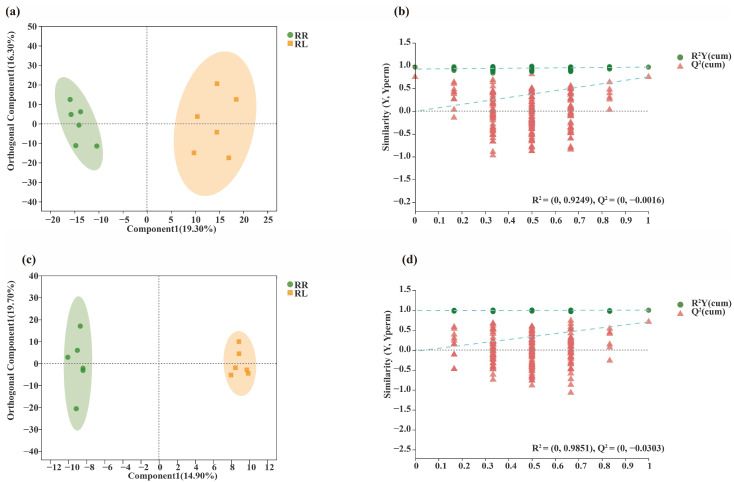
Orthogonal partial least squares discriminant analysis (OPLS-DA) plot of RR and RL metabolites when comparing the RR and RL groups following (**a**) positive mode ionization; (**b**) permutation test of OPLS-DA in positive mode ionization; (**c**) negative mode ionization; (**d**) permutation test of OPLS-DA in negative mode ionization.

**Figure 10 microorganisms-11-00326-f010:**
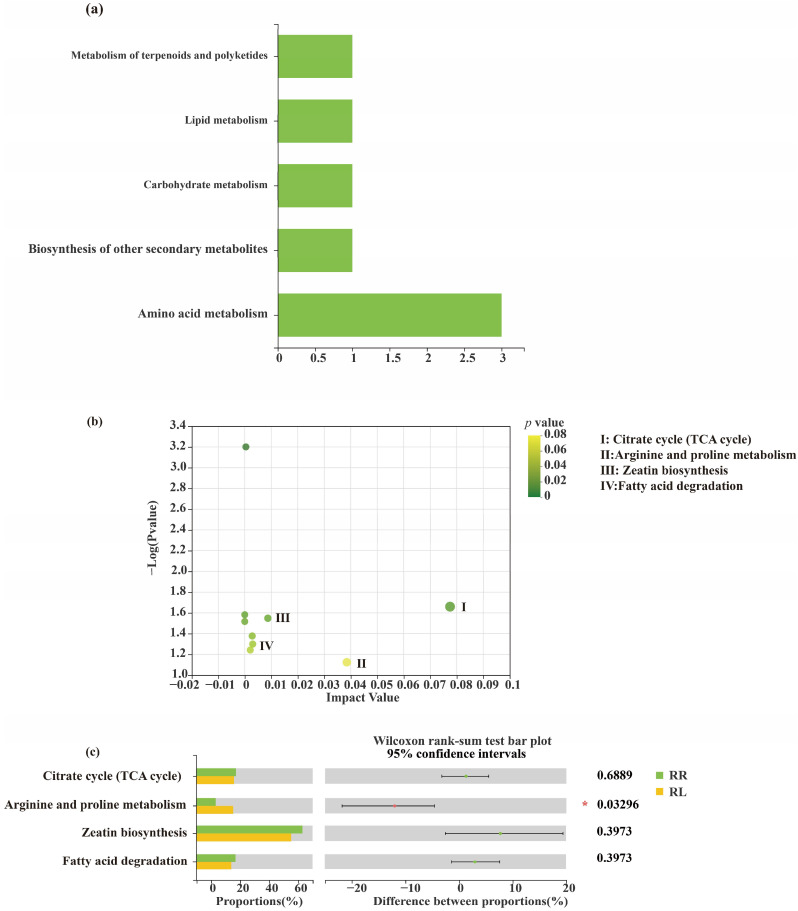
(**a**) KEGG pathway classification: metabolites detected and annotated. (**b**) Metabolic pathway enrichment study of differentially abundant metabolites between the oilseed rape monoculture (RR), intercropping system (RL). (**c**) The changes in root exudates in different metabolic pathways by KEGG. * The representation is significant.

**Figure 11 microorganisms-11-00326-f011:**
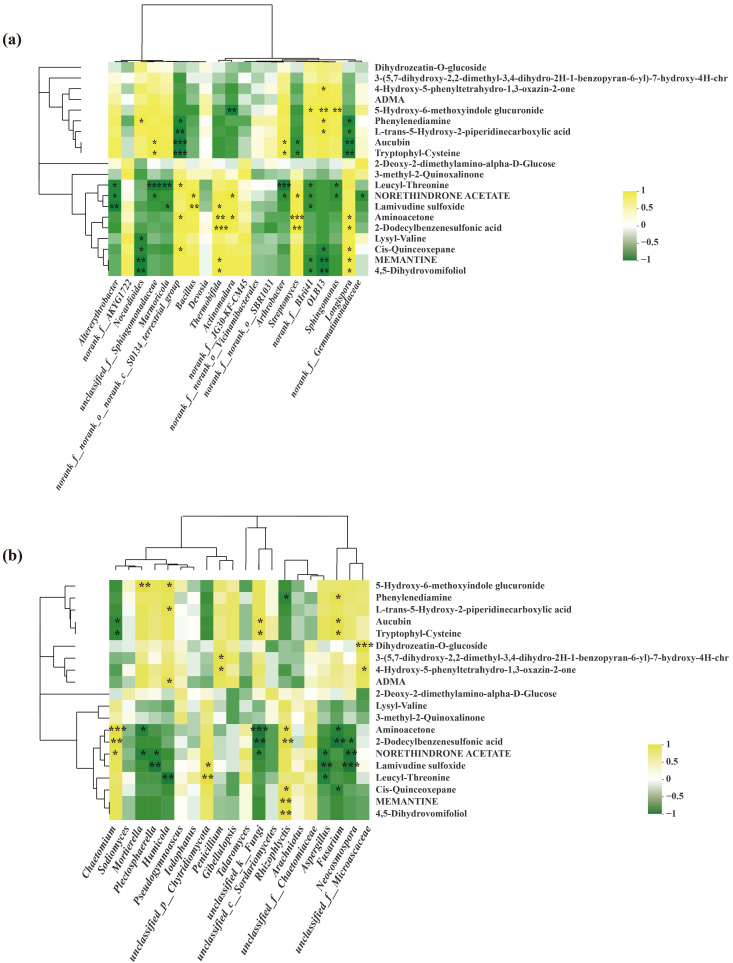
Correlation of metabolites in root exudates and bacterial (**a**) and fungal (**b**) communities in rhizospheres of oilseed rapes under RR and RL systems. Heatmaps analyze the Spearman correlation coefficient (rho value) and *p* value of metabolites and flora. * 0.01 < *p* < 0.05, ** 0.001 < *p* < 0.01, *** *p* < 0.001. This indicates that the difference is significant. The clustering at the top and left of the graph shows the metabolites of the bacterial groups and the results of hierarchical clustering based on Euclidean distance, respectively.

**Table 1 microorganisms-11-00326-t001:** Sequencing type and primer sequence.

Sequencing Type	Primer Name	Primer Sequence	Length	Sequencing Platform
Bacterial 16SrRNA	338F	5’-ACTCCTACGGGAGGCAGCAG-3’	311 bp	MiSeq PE250
	806R	5’-GGACTACHVGGGTWTCTAAT-3’		
Fungal ITS	ITS1F	5’-CTTGGTCATTTAGAGAAGTAA-3’	350 bp	MiSeq PE300
	ITS2F	5’-GCTGCGTTCTTCATCGATGC-3’		

**Table 2 microorganisms-11-00326-t002:** Gradient of mobile phases.

Time (min)	Flow Rate (mL/min)	A (%)	B (%)
0	0.4	100	0
0.1	0.4	95	0
2	0.4	75	25
9	0.4	0	100
13	0.4	0	100
13.1	0.4	100	0

**Table 3 microorganisms-11-00326-t003:** Soil bacterial alpha diversity in rhizospheres of oilseed rape under RR, RL and CK systems.

	Shannon Index	Insimpson Index	Ace Index	Chao1 Index	Coverage
RR	6.58 ± 0.05 a	270.42 ± 13.98 a	3544.93 ± 25.42 a	3562.24 ± 37.55 a	0.97
RL	6.49 ± 0.04 ab	234.76 ± 13.12 b	3492.19 ± 92.62 a	3490.77 ± 95.95 a	0.97
CK	6.46 ± 0.05 b	224.18 ± 20.39 b	3581.83 ± 47.9 a	3532.32 ± 111.15 a	0.97

Note: Data in the table are means ± SDs. Values followed by different lowercase letters indicate significant differences between different rows (*p* < 0.05).

**Table 4 microorganisms-11-00326-t004:** Soil fungal alpha diversity in rhizospheres of oilseed rape under RR, RL and CK systems.

	Shannon Index	Insimpson Index	Ace Index	Chao1 Index	Coverage
RR	3.7 ± 0.07 a	18.22 ± 2.45 a	696.95 ± 11.3 a	671.84 ± 8.13 a	0.99
RL	3.52 ± 0.10 b	13.18 ± 1.25 b	660.93 ± 15.24 a	669.31 ± 30.98 a	0.99
CK	3.45 ± 0.07 b	13.61 ± 2.08 b	1656.28 ± 42.63 a	634.92 ± 41.71 a	0.99

Note: Data in the table are means ± SDs. Values followed by different lowercase letters indicate significant differences between different rows (*p* < 0.05).

**Table 5 microorganisms-11-00326-t005:** The metabolites of differential root exudates in rhizospheres of oilseed rapes between RR and RL systems.

Metabolite	RR	RL	RR/RL	*p*-Value
Organic acids and their derivatives				
N-Undecanoylglycine	5.86 ± 0.02 b	5.97 ± 0.11 a	0.98	0.0433
Citric acid	3.86 ± 0.15 a	3.69 ± 0.07 b	1.05	0.0299
Creatine	3.08 ± 0.12 b	3.59 ± 0.33 a	0.86	0.0051
ADMA	4.37 ± 0.12 a	4.15 ± 0.21 b	1.05	0.0471
Tryptophyl-cysteine	4.37 ± 0.14 a	3.79 ± 0.50 b	1.15	0.0208
Isoleucylproline	3.00 ± 0.14 b	3.48 ± 0.49 a	0.86	0.0436
2-Amino-4-[(2-hydroxy-1-oxopropyl)amino] butanoic acid	3.99 ± 0.25 a	3.58 ± 0.25 b	1.11	0.0183
L-trans-5-Hydroxy-2-piperidinecarboxylic acid	4.37 ± 0.22 a	4.03 ± 0.22 b	1.09	0.0225
Lysyl-valine	4.33 ± 0.03 b	4.42 ± 0.06 a	0.98	0.0098
Leucyl-threonine	4.35 ± 0.05 b	4.61 ± 0.03 a	0.94	0.000001
Organic oxygen compounds				
5-Hydroxy-6-methoxyindole glucuronide	3.86 ± 0.10 a	3.63 ± 0.11 b	1.25	0.0248
Aminoacetone	4.89 ± 0.03 b	4.99 ± 0.04 a	0.98	0.0008
Organic nitrogen compounds				
L-Carnitine	3.01 ± 0.1 b	3.42 ± 0.33 a	0.88	0.0163
Memantine	4.72 ± 0.02 b	4.86 ± 0.12 a	0.97	0.0229
Tromethamine	2.56 ± 0.69 b	3.32 ± 0.38 a	0.77	0.0410
Organoheterocyclic compounds				
Cis-quinceoxepane	4.95 ± 0.03 b	5.09 ± 0.12 a	0.97	0.0216
9-Azabicyclo [3.3.1] nonan-3-one	3.01 ± 0.05 b	3.66 ± 0.64 a	0.82	0.0312
5-Oxo-2(5H)-isoxazolepropanenitrile	2.98 ± 0.21 b	3.29 ± 0.26 a	0.91	0.0449
Phenylpropanoids and polyketides				
2-Pentyl-3-phenyl-2-propenal	2.66 ± 0.32 b	3.68 ± 0.99 a	0.72	0.0363
Nucleosides, nucleotides and their analogs				
Lamivudine sulfoxide	4.95 ± 0.04 b	5.04 ± 0.02 a	0.98	0.0010
7-Methylinosine	3.87 ± 0.15 a	3.63 ± 0.17 b	1.07	0.0288
Lipids and lipid-like molecules				
Aucubin	4.43 ± 0.18 a	3.84 ± 0.52 b	1.15	0.0266
Etonogestrel	2.74 ± 0.04 a	2.61 ± 0.08 b	1.05	0.0034
4,5-Dihydrovomifoliol	4.35 ± 0.02 b	4.5 ± 0.14 a	0.97	0.0346
(2S,4R)-p-Mentha-1(7),8-dien-2-ol	3.3 ± 0.06 b	3.55 ± 0.22 a	0.93	0.0233
Dihydrozeatin-O-glucoside	4.45 ± 0.19 a	4.23 ± 0.13 b	1.05	0.0419
Benzenoids				
Benzaldehyde	4.62 ± 0.03 b	4.71 ± 0.05 a	0.98	0.0061
4-Hydroxy-5-phenyltetrahydro-1,3-oxazin-2-one	4.36 ± 0.06 a	4.24 ± 0.05 b	1.03	0.0023
2-Dodecylbenzenesulfonic acid	4.82 ± 0.05 b	4.99 ± 0.03 a	0.97	0.000021
Other				
Norethindrone Acetate	4.99 ± 0.06 b	5.12 ± 0.02 a	0.97	0.0006
3-methyl-2-Quinoxalinone	3.66 ± 0.1 b	4.36 ± 0.43 a	0.84	0.0030
13-Hydroxylupanine	2.95 ± 0.14 a	2.59 ± 0.32 b	1.18	0.0323
Allothreonine	4.11 ± 0.05 b	4.20 ± 0.05 a	0.98	0.0048
Eicosanoyl-EA	3.5 ± 0.10 a	3.10 ± 0.32 b	1.13	0.0165
5-Hydroxyindol-2-carboxylic acid	3.86 ± 0.10 a	3.63 ± 0.11 b	1.06	0.0035
3,5-dihydroxy-2-(3-hydroxyphenyl)-8,8-dimethyl-4H,8H-pyrano[2,3-f] chromen-4-one	3.77 ± 0.41 a	2.73 ± 0.66 b	1.38	0.0085
3,4,5-trihydroxy-6-[(2-phenylacetyl)oxy] oxane-2-carboxylic acid	3.9 ± 0.31 a	3.52 ± 0.28 b	1.11	0.0471
2-Deoxy-2-dimethylamino-alpha-D-Glucose	3.67 ± 0.43 b	4.39 ± 0.60 a	0.84	0.0363
2-AI	2.90 ± 0.19 b	3.91 ± 0.30 a	0.74	0.00004
3-(5,7-dihydroxy-2,2-dimethyl-3,4-dihydro-2H-1-benzopyran-6-yl)-7-hydroxy-4H-chromen-4-one	4.21 ± 0.18 a	3.66 ± 0.47 b	1.15	0.0217
8-(1,2-dihydroxypropan-2-yl)-9-hydroxy-2H,8H,9H-furo[2,3-h] chromen-2-one	3.74 ± 0.19 a	3.19 ± 0.3 b	1.17	0.0037
Phenylenediamine	4.63 ± 0.18 a	4.40 ± 0.17 b	1.05	0.0472

Note: Data in the table are means ± SDs. Values followed by different lowercase letters indicate significant differences between different rows (*p* < 0.05).

## Data Availability

Raw data for rhizospheric bacterial and fungal sequencing were deposited in the NCBI Sequence Read Archive (SRA) database under accession number PRJNA912616 (accessed on 15 December 2022) and PRJNA912881 (accessed on 16 December 2022), respectively.

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
