# Peer review of "Changes in Soil Phosphorus Availability and Microbial Community Structures in Rhizospheres of Oilseed Rapes Induced by Intercropping with White Lupins"

_microorganisms, 2023, doi:10.3390/microorganisms11020326_

Round 1

Reviewer 1 Report

Brief Summary

The manuscript microorganisms-2134425 investigated the soil fertility, bacterial and fungal communities of rhizosphere and metabolites of oilseed rape root exudates in monoculture and intercropping systems with wite lupins under a greenhouse experiment.

The manuscript contains interesting findings. However, the manuscript presents major flaws in the quality of the presentation. I provided my specific comments in the pdf file attached.

Specific comments

·         Introduction: The introduction does not place the study in a broad context or highlight why the study was needed. The working hypotheses must be presented.

·         Materials and Methods: The description of the methods lacks some details. See the comments on the pdf file attached.

·         Results: The results section is not clear. Even if the quality of the figure is satisfactory the results description is confusing. The authors should present clearly the main findings to support their results. The quality of the presentation must be revised for this section. The tables’ titles and figures’ captions must be clear and the content double checked before resubmission. See the comments on the pdf file attached.

·         Discussion: The treatment is not well structured and does not present a clear flow of concepts. The list of references needs to be enriched.

·         Conclusions: Beyond a summary of the study the authors should underline the importance of the findings obtained and the future research directions.

Other comments

The English language must be revised throughout the manuscript for fluency and correctness.

The italics of the genera must be also revised.

Author Response

Response to Reviewer Comments

Brief Summary

The manuscript microorganisms-2134425 investigated the soil fertility, bacterial and fungal communities of rhizosphere and metabolites of oilseed rape root exudates in monoculture and intercropping systems with white lupins under a greenhouse experiment.

The manuscript contains interesting findings. However, the manuscript presents major flaws in the quality of the presentation. I provided my specific comments in the pdf file attached.

Specific comments

Introduction: The introduction does not place the study in a broad context or highlight why the study was needed. The working hypotheses must be presented.

Response 1: The authors have revised this part in introduction according to the Reviewer’s suggestion in manuscript (L79-82).

Materials and Methods: The description of the methods lacks some details. See the comments on the pdf file attached.

Response 2: The authors have revised this part in materials and methods according to the Reviewer’s suggestion in manuscript.

Results: The results section is not clear. Even if the quality of the figure is satisfactory the results description is confusing. The authors should present clearly the main findings to support their results. The quality of the presentation must be revised for this section. The tables’ titles and figures’ captions must be clear and the content double checked before resubmission. See the comments on the pdf file attached.

Response 3: The authors have revised this part in materials and methods according to the Reviewer’s suggestion in manuscript.

Discussion: The treatment is not well structured and does not present a clear flow of concepts. The list of references needs to be enriched.

Response 4: The authors have revised this part in materials and methods according to the Reviewer’s suggestion in manuscript.

Conclusions: Beyond a summary of the study the authors should underline the importance of the findings obtained and the future research directions.

Response 5: The authors have revised this part in materials and methods according to the Reviewer’s suggestion in manuscript.

Other comments

The English language must be revised throughout the manuscript for fluency and correctness.

Response 6: The authors have revised this point in manuscript according to the Reviewer’s suggestion.

The italics of the genera must be also revised.

Response7: The authors have revised this point in manuscript according to the Reviewer’s suggestion.

L78-L80 Missing working hypotheses.

Response 8: The authors have revised this point in manuscript according to the Reviewer’s suggestion (L79-82).

L82 For each experiment the authors should add the number of biological replicates analyzed.

Response 9: The authors have revised this point in manuscript according to the Reviewer’s suggestion (L93-94).

L98 identify these stages as days after sowing and using a BBCH scale code.

Response 10: The authors have revised this point in manuscript according to the Reviewer’s suggestion (L100-101).

L100 identify it as days after sowing and BBCH scale code.

Response 11: The authors have revised this point in manuscript according to the Reviewer’s suggestion (L102-103).

L101 Add the total number of plants.

Response 12: The authors have revised this point in manuscript according to the Reviewer’s suggestion (L104).

L107 “samples” changed to Soil samples.

Response 13: The authors have revised this point in manuscript according to the Reviewer’s suggestion (L106).

L111 missing reference, revise this sentence.

Response 14: The authors have added the reference in manuscript according to the Reviewer’s suggestion (L110).

L115 with 200 mL of sterile water?

Response 15: The authors have revised this point in manuscript according to the Reviewer’s suggestion (L118-120).

L116 “for continuous aeration” not clear.

Response 16: The authors have revised this point in manuscript according to the Reviewer’s suggestion (L113-115).

L122 Describe breifly the method.

Response 17: The authors have revised this point in manuscript according to the Reviewer’s suggestion (L121-125).

L153 Sequencing data processing workflow is missing. Authors should add bioinformatic and biostatistic approaches used.

Response 18: The authors have revised this point in manuscript according to the Reviewer’s suggestion (L146-167).

L156-157 Revise the sentence.

Response 19: The authors have revised this point in manuscript according to the Reviewer’s suggestion (L170-179).

L195-196 The discussion of the results should be avoided in the results section.

Response 20: The authors have deleted this point in manuscript according to the Reviewer’s suggestion.

L202 “in the same column” Revise it.

Response 21: The authors have revised this point in manuscript according to the Reviewer’s suggestion (L233-235).

L212 “Simpson index“  The values are very low. Based on the values calculated for the other diversity indices, the results should be higher. The Methods section is deficient in the results of the index calculation. I suggest revising these results. Index calculation procedures should be added in the methods section.

Response 22: The authors have revised this point in manuscript according to the Reviewer’s suggestion (L236-249).

L216 Beyond the mistake of the missing table (table 3 was presented twice), it is not clear the content of this part of the results presentation.

Response 23: The authors have revised this point in manuscript according to the Reviewer’s suggestion (L236-249).

L233 The discussion of the results should be avoided in the results section.

Response 24: The authors have deleted this point in manuscript according to the Reviewer’s suggestion.

L261 “Proportions” Revise the term with an appropriate one.

Response 25: The authors have revised this point in manuscript according to the Reviewer’s suggestion (L281-282).

L263-291 This part of the description of the results should be completely revised for clarity. the authors should present the most significant results and highlight the most important aspects.

Response 26: The authors have revised this point in manuscript according to the Reviewer’s suggestion (L283-302).

L304-310 The discussion of the results should be avoided in the results section.

Response 27: The authors have deleted this point in manuscript according to the Reviewer’s suggestion.

L384-385 Revise the caption for clarity.

Response 28: The authors have revised this point in manuscript according to the Reviewer’s suggestion (L410-413).

L394 The title of the table needs revision.

Response 29: The authors have revised this point in manuscript according to the Reviewer’s suggestion (L422-423).

L394 “RR/RL“ The metabolite concentrations should be presented with their means and standard deviations before presenting their ratios.

Response 30: The authors have added means and standard deviations in manuscript according to the Reviewer’s suggestion (L423).

L394 “Lon (M/Z) “  These data could be presented in a table in a supplementary material file and presented in the material and methods section.

Response 31: The authors have revised this point in manuscript according to the Reviewer’s suggestion (L423).

L394 “p < 0.05.” This data is not useful, you should present the p value. It is not clear the statistical test performed.

Response 32: The authors have revised this point in manuscript according to the Reviewer’s suggestion (L423). The screening criteria were VIP_pred_OPLS-DA > 1, P-value < 0.05.

L396 How were KEGG pathways estimated?

Response 33: Metabolites in the metabolic set were classified according to the pathway pathway they participate in or the function they exercise using the KEGG database (KyotoEncyclopediaofGenesandGenomes, http://www.genome.jp/kegg). The metabolites in the metabolic set are displayed on a KEGGpathway pathway map, showing a map of the KEGG annotated pathways in which they are involved. The metabolites in the metabolic set were subjected to KEGGPATHWAY enrichment analysis using Fisher's exact test for enrichment analysis. The p-value was also calibrated using BH (BenjaminiandHochberg) with a threshold of 0.05, and KEGG pathways meeting this condition were defined as significantly enriched KEGG pathways in the metabolic set.

L410-473 This description is not appropriate.

Response 34: The authors have revised this point in manuscript according to the Reviewer’s suggestion (L439-470).

L481 The discussion section needs substantial revisions. The authors should discuss the findings in the perspective of previous studies and the working hyphotheses with a clear folw of concepts. Only few references of 2020 and 2021 were considered. No references of 2022 were considered.

Response 35: The authors have found and added the refernces below in this manuscript. i.e. Zhang et al., 2023(48); Aslam et al., 2022 (49), Boudsocq et al., 2022 (50); Schwerdtner and Spohn, 2022 (51); 5Aslam et al., 2022 (58) to support the statements and revised this point in manuscript according to the Reviewer’s suggestion (L478-542).

L556 除了对结果的总结外,作者还应解释所获得结果的重要性以及它们如何丰富该领域的知识。此外,应提及未来的研究方向,描述验证所获得结果的步骤

回应36:作者根据审稿人的建议(L544-555)在稿件中修改了这一点。

特别感谢您的好评。

Reviewer 2 Report

Manuscript title: Changes of soil phosphorus availability and microbial community structures in rhizospheres of oilseed rapes by intercropping with white lupins   

Manuscript id: microorganisms-2134425

Authors: Chen et al.

The manuscript is particularly strong regarding the less studied topic and the experimental setup on intercropping oilseed rapes with white lupins……. The manuscript regarding the topic and results presented is of interest to plant science community and revisions based on the comments below are recommended before considering for publication.

Major comments

·       Insufficient Abstract: In the abstract, the main aim and background of the manuscript are missing, the current version it only highlights the result. In addition, it would be even better to have a sentence as a future perspective.

·       The unit/abbreviation is not mentioned before, consider defining the abbreviation when mentioned for the first time…. Please check throughout the manuscript to define the abbreviations.

·       Line 78-81, the aim or hypothesis of the study is clear, however, the approach is missing ….

·       Lake of scientific literature to support the statements and findings throughout the manuscript…... I have made some suggestions for that and more need it….

·       More information is needed for ALL TABLE captions and define the abbreviation and units that are used. And adjust the significant figures for the table and manuscript.

·       Grammar and punctuation issuers need to be addressed. I have selected/mentioned some as examples.

·       I have a major concern about the results and discussion section. The authors describe the results and compare the results with previous studies, however, insight mechanisms are still insufficient.

Detailed comments:

Abstract

If the unit/abbreviation is not mentioned before, consider defining the abbreviation when mentioned for the first time.

Line 16-18: A complicated sentence

Line 23-28: These are rather long sentences, better to break them down into more sentences.

Introduction:

Line 39: A reference is needed here, for example, you can use:

Line 52-57: A complicated sentence, please revise and check the grammar

Line 60: A reference is needed here, for example, you can use:

https://doi.org/10.1186/s12302-020-00405-7

Line 68-78: These are rather long sentences, better to break them down into more sentences.

In MM section

Literature references are missing for all sub-section. It would be better to cite the references that the procedure adopted.

Additional info is needed for the table caption, most importantly significant figures.

In MM section, what is the quality control (QC) data? There is no mention of the QC.

What is the accuracy of the instruments, recovery, LOD, and LOQ ……. These parameters are needed to report the efficiency of any analytical system.

In general, how many times you’ve recorded the data,? duplicate? Triplicate?..... what you mentioned in the text is not clear, please elaborate more on this

R&D section

Figure 1. how the comparison made between the treatments and assigning the letter for the statistical difference is confusing. For example, How you can have a, b; ab? Please elaborate more….

These sections are repeating information already presented and explain things in an unnecessarily complicated way. The quality of the manuscript would benefit from the whole section being condensed, line 263-290,  line 370-382, line 413-446, line 505-555…

Conclusion

Important conclusions! However, the future perspectives for the following research are highly crucial here …..

Author Response

回复审稿人 评论

手稿特别强烈地涉及研究较少的主题和将油菜与白羽扇豆间作的实验装置......植物科学界对所介绍的主题和结果感兴趣的手稿,建议在考虑发表之前根据以下评论进行修订。

主要评论

摘要不足:在摘要中,缺少稿件的主要目的和背景,当前版本仅突出结果。此外,最好有一个句子作为未来的观点。

回应1:作者根据审稿人的建议(L12-18)修改了稿件中的这一点。

单位/缩写之前没有提到,考虑在第一次提到时定义缩写。请检查整个手稿以定义缩写。

回应2:作者根据审稿人的建议,在稿件中修改了这一点。

第 78-81 行,研究的目的或假设很明确,但是缺少该方法......

回应3:作者根据审稿人的建议(L79-82)在稿件中修改了这一点。

湖科学文献支持整个手稿的陈述和发现......我为此提出了一些建议,更需要它......

回应4作者在本手稿中找到并添加了以下参考文献。即阿马杜等人,2021 (5);哈马等人,2020 (29);博特彻等人,2016 (40);扎尔尼娜等人,2018 (42);陈等人, 2022 (45);陈等, 2022 (46);张等, 2023(48);阿斯拉姆等人,2022 (49),布德索克等人,2022 (50);施韦特纳和斯波恩,2022 (51);阿斯拉姆等人,2022 (58);以支持这些陈述。

所有表格标题都需要更多信息,并定义使用的缩写和单位。并调整表格和手稿的有效数字。

回应5:作者根据审稿人的建议在稿件中修改了这一点。

需要解决语法和标点符号发布者的问题。我选择/提到了一些作为例子。

回应6:作者根据审稿人的建议修改了稿件中的这一点。

我对结果和讨论部分非常关注。作者描述了结果并将结果与以前的研究进行了比较,然而,洞察力机制仍然不足。

回应7:作者根据审稿人的建议(L478-542)修改了稿件中的这一点。

详细评论:

抽象

如果之前未提及单位/缩写,请考虑在首次提及时定义缩写。

回应8:作者根据审稿人的建议修改了稿件中的这一点。

第16-18行:一个复杂的句子

回应9:作者根据审稿人的建议(L15-18)在稿件中修改了这一点。

第23-28行:这些是相当长的句子,最好将它们分解成更多的句子。

回应10:作者根据审稿人的建议(L23-27)在稿件中修改了这一点。

介绍:

第 39 行:这里需要一个参考,例如,您可以使用:

Response 11 : The authors have added Amadou et al., 2021 (5) in References according to the Reviewer’s suggestion in manuscript

Line 52-57: A complicated sentence, please revise and check the grammar

Response 12: The authors have revised this point in manuscript according to the Reviewer’s suggestion (L51-54).

Line 60: A reference is needed here, for example, you can use:https://doi.org/10.1186/s12302-020-00405-7

Response 13: The authors have added Hama, J.R. and Strobel, B.W., 2020 (29) in References according to the Reviewer’s suggestion in manuscript.

Line 68-78: These are rather long sentences, better to break them down into more sentences.

Response 14: The authors have revised this point in manuscript according to the Reviewer’s suggestion (L68-77).

In MM section

Literature references are missing for all sub-section. It would be better to cite the references that the procedure adopted.

Response 15: The authors have added Böttcher et al., 2016 (40); Zhalnina et al., 2018 (42); Chen et al., 2022 (45); Chen et al., 2022 (46) in References according to the Reviewer’s suggestion in manuscript

Additional info is needed for the table caption, most importantly significant figures.

In MM section, what is the quality control (QC) data? There is no mention of the QC.

Response 16: The authors have revised this point in manuscript according to the Reviewer’s suggestion (L175-179).

What is the accuracy of the instruments, recovery, LOD, and LOQ ……. These parameters are needed to report the efficiency of any analytical system.

Response 17: The authors have revised this point in manuscript according to the Reviewer’s suggestion (L146-167).

一般来说,您记录了多少次数据?重复?三重?。。。。。你在文中提到的不清楚,请详细说明

回应18:作者根据审稿人的建议(L93,L103-104)在稿件中修改了这一点。

图1.处理之间的比较和为统计差异分配字母的方式令人困惑。例如,你怎么能有a,b;血型?请详细说明更多...

响应 19:使用 Excel 2019 和统计产品和服务解决方案 (SPSS) 21 对试验数据进行统计分析,结果显示为具有标准差(平均值± SD)的均值。使用SPSS进行单因素检验,计算显著性,并在图中执行

可用

土壤磷酸酶

活动

土壤微生物

生物质磷

RR

16.64±0.15字节

2.90±0.30安培

26.14±6.93一分

RL

16.83±0.06安

2.74±0.15一分

31.33±7.87一

CK

16.49±0.08字节

2.30±0.18乙

16.07±6.95字节

这些部分重复已经提供的信息,并以不必要的复杂方式解释事物。手稿的质量将受益于整个部分的压缩,第 263-290 行、370-382 行、413-446 行、505-555 行......

回应20:作者根据审稿人的建议(L283-302,L402-408,L439-470,L479-542)修改了稿件中的这一点。

结论

重要结论!然而,以下研究的未来前景在这里非常重要.....

回应21:作者根据审稿人的建议(L544-555)修改了手稿中的这一点。

特别感谢您的好评。

Round 2

Reviewer 1 Report

The authors correctly addressed all my previous suggestions. I have no further comments.

Reviewer 2 Report

The revised manuscript has improved compared to the original version. The authors tried to address my questions as much as possible. I recommend the manuscript to be published!